# Noise-Induced Hearing Loss: Overview and Future Prospects for Research on Oxidative Stress

**DOI:** 10.3390/ijms26104927

**Published:** 2025-05-21

**Authors:** Tsubasa Kitama, Takanori Nishiyama, Makoto Hosoya, Marie N. Shimanuki, Masafumi Ueno, Fukka You, Hiroyuki Ozawa, Naoki Oishi

**Affiliations:** 1Department of Otorhinolaryngology Head and Neck Surgery, Keio University School of Medicine, 35 Shinanomachi, Shinjuku-ku, Tokyo 160-8582, Japan; tsubasa.kitama@gmail.com (T.K.); mhosoya1985@gmail.com (M.H.); nukimari@gmail.com (M.N.S.); uenomasafumi0706@yahoo.co.jp (M.U.); ozakkyy123@gmail.com (H.O.); oishin@keio.jp (N.O.); 2Division of Anti-Oxidant Research, Life Science Research Center, Gifu University, Gifu 501-1193, Japan; y@antioxidantres.jp; 3Anti-Oxidant Research Laboratory, Louis Pasteur Center for Medical Research, Kyoto 606-8225, Japan

**Keywords:** reactive oxygen species (ROS), acoustic trauma, antioxidant

## Abstract

Noise-induced hearing loss (NIHL) is a common type of sensorineural hearing loss caused by exposure to high-intensity noise that leads to irreversible cochlear damage. Despite extensive research on cochlear pathophysiology, the precise mechanisms remain unclear, and no established treatment exists. This is due to the challenges in imaging and the inability to perform biopsies in human patients. Consequently, animal models, particularly mice, have been widely used to study NIHL. Clinically, NIHL presents as either a temporary threshold shift, in which hearing recovers, or a permanent threshold shift, which results in an irreversible loss. Histopathological studies have identified the key features of NIHL, including outer hair cell loss, auditory nerve degeneration, and synaptic impairment. Recent findings suggest that oxidative stress and inflammation are major contributors to NIHL, highlighting the potential for therapeutic interventions, such as antioxidants and anti-inflammatory agents. Given the increasing prevalence of NIHL owing to occupational noise exposure and personal audio device use, addressing this issue is a pressing public health challenge. This review summarizes the clinical features, underlying mechanisms, and emerging treatment strategies for NIHL while identifying current knowledge gaps and future research directions.

## 1. Overview of Noise-Induced Hearing Loss

Noise-induced hearing loss (NIHL) is a type of sensorineural hearing loss resulting from prolonged exposure to high noise levels, leading to irreversible damage to hair cells in the cochlea [1]. NIHL can be categorized as acute or chronic. Previous results from a single exposure to intense noise, such as gunshot noise, can cause immediate hearing impairment [2]. The latter develops gradually because of continuous exposure to loud sounds over time and is commonly associated with occupational or recreational activities [3]. Although various standards and measures have been established in workplaces, the prevalence of NIHL remains high. NIHL is the second most common cause of sensorineural hearing loss after age-related hearing loss, and approximately 5% of the global population has NIHL [4]. The economic impact of occupational hearing loss is projected to reach billions of dollars [5].

The advancement of audio technology and the rapid expansion of the music industry have contributed to a growing trend among adolescents and young adults of voluntarily listening to music at high volumes for extended periods. Such prolonged exposure to loud sounds significantly increases the risk of NIHL, raising public health concerns about a potential rise in the prevalence of NIHL in the future [6]. The social implications of NIHL are profound, affecting communication abilities, social interactions, and overall quality of life. Several studies have reported that hearing loss is associated with both dementia [7,8] and depression [9]. Given the anticipated increase in the number of individuals with hearing loss owing to aging and the rising prevalence of earphone-induced hearing loss, addressing this issue represents a socially significant challenge that requires urgent attention.

NIHL is a common form of acute sensorineural hearing loss. Although the cochlear pathophysiology underlying NIHL has been extensively studied, the exact mechanisms are not fully understood. Consequently, no established treatment has been developed. This limitation arises from two major challenges: the small size of the cochlea, which makes detailed imaging evaluations difficult, and the inability to perform biopsies in patients with HL. Consequently, much research on hearing loss has relied on animal models.

One of the most widely used models for studying acute sensorineural hearing loss is the noise-induced trauma mouse model. This model is commonly employed owing to its relative cost-effectiveness and simplicity of creation [10,11]. The increasing prevalence of noise exposure from earphones and other personal audio devices suggests that NIHL cases will continue to increase, potentially becoming a significant public health concern [12]. Noise exposure-related auditory damage has been extensively characterized in animal studies, providing a robust foundation for future research.

This noise-induced trauma model offers valuable insights into the clinical progression and underlying mechanisms of NIHL. Histological analysis often reveals damage to the outer hair cells, inner hair cells, and synapses connecting these cells to spiral ganglion neurons. Recent studies have identified oxidative stress and inflammation as critical contributors to NIHL [13,14]. These findings underscore the importance of targeted therapeutic strategies such as antioxidants and anti-inflammatory agents, which are being explored in preclinical models.

In this review, we summarize the clinical course, underlying mechanisms, and potential therapeutic approaches of NIHL. Using recent publications, we highlight the current gaps in knowledge and outline future research directions.

## 2. Clinical Features of NIHL

Hearing loss caused by noise exposure can recover to baseline levels over several hours to weeks after exposure; this is referred to as a temporary threshold shift (TTS) [15]. However, when the intensity of the noise is higher or the duration of exposure is longer, permanent hearing loss may persist, known as permanent threshold shift (PTS) [16]. Animal studies indicate that immediate threshold shifts of approximately 50 dB or less may result in complete recovery, whereas shifts exceeding this level are more likely to result in permanent hearing loss [17,18]. Additionally, chronic exposure to noise levels of 85 dB or higher for 8 h per day can cause permanent hearing loss [19].

Although hearing thresholds recover after TTS, the reduced ability to understand speech in noisy environments may remain impaired [20]. This residual damage, known as hidden hearing loss, can occur even when TTS appears to have resolved. Repeated exposure to noise levels that initially cause TTS can lead to PTS [21]. Therefore, noise exposure that causes only TTS should not be ignored as it may result in cumulative damage and progression to PTS.

## 3. Histopathological Changes in the Cochlea in NIHL

Collecting human inner ear tissue to observe histological changes in NIHL is challenging. Therefore, various animal models have been used to elucidate the mechanisms underlying NIHL, including mice [22], rats [23], chinchillas [24], zebrafish [25], guinea pigs [26], and nonhuman primates [27].

By varying the sound intensity, duration, and frequency bands, it is possible to create animal models of NIHL involving either PTS or TTS. Figure 1 shows the histological changes associated with PTS and TTS which have been identified.

### 3.1. PTS

(i)Hair Cell Death

Prolonged or intense noise exposure induces apoptosis or necrosis of outer hair cells, leading to irreversible hearing loss. The loss of hair cells impairs their ability to convert sound vibrations into neural signals [28].

(ii)Degeneration of Auditory Nerve Fibers

Following hair cell loss, spiral ganglion neurons undergo degeneration owing to the loss of trophic support from hair cells, further exacerbating hearing impairment [28].

(iii)Damage to Supporting Cells and Cochlear Structures

Severe noise exposure damages the supporting cells, stria vascularis, and other cochlear structures. This damage disrupts ion homeostasis and cochlear stability, contributing to permanent auditory dysfunction [28].

### 3.2. TTS

In cases of TTS, no significant histological changes, such as those observed in PTS, are observed [29]. However, the following changes occur:(i)Synaptic Damage

In TTS, hair cells remain intact, but the ribbon synapses between inner hair cells (IHCs) and cochlear nerve fibers are impaired [29,30].

(ii)Buckling of Supporting Cells

Moderate noise exposure can induce buckling of supporting cells, such as pillar cells, resulting in the detachment of outer hair cell (OHC) stereocilia from the tectorial membrane, which is believed to cause TTS. The mechanism underlying TTS appears to be distinct from those responsible for permanent hair cell damage and PTS [28].

## 4. Molecular Mechanisms Underlying NIHL

Mechanisms underlying NIHL are complex and multifaceted. Acoustic trauma caused by exposure to intense sound affects various cellular components of the inner ear, particularly the hair cells, auditory neurons, and surrounding supporting cells. NIHL pathogenesis involves a combination of interrelated factors, including oxidative stress, calcium dysregulation, mitochondrial dysfunction, inflammatory responses, and direct mechanical damage induced by acoustic overstimulation. Collectively, these processes contribute to the auditory dysfunction and hearing loss.

### 4.1. Oxidative Stress

Oxidative stress is one of the most crucial factors in NIHL. Exposure to acoustic stimulation induces excessive production of reactive oxygen species (ROS) in the inner ear, which damages cellular structures and functions [31]. Among the ROS, the superoxide anion (O_2_^−^), hydroxyl radical (·OH), and hydrogen peroxide (H_2_O_2_) have been frequently reported to accumulate in the cochlea following acoustic overstimulation [32,33,34]. Similarly, reactive nitrogen species such as nitric oxide (NO) and peroxynitrite (ONOO^−^) are also generated in response to noise exposure [35]. ROS oxidizes lipids in cell membranes, DNA, and proteins, leading to cellular degeneration and destruction. Mitochondrial DNA (mtDNA) is particularly susceptible to damage owing to the lack of protection by histone proteins and limited DNA repair mechanisms. Consequently, these cells are highly prone to mitochondrial dysfunction. Mitochondria are energy-producing centers that supply energy to inner ear cells through ATP synthesis. However, when ROS levels increase after NIHL, mitochondrial membranes are damaged, and ATP synthesis is disrupted. Consequently, there is an energy deficit within the cells, ultimately leading to cell death. Furthermore, mitochondrial damage exacerbates calcium stress and accelerates cellular damage following NIHL [36,37]. The inner ear, particularly hair cells, has a high metabolic demand and consumes significant amounts of oxygen, making it highly susceptible to ROS generation [33]. In addition, damage to supporting cells, the stria vascularis, and auditory nerve cells in the inner ear can ultimately lead to hearing loss [38].

Prolonged oxidative stress induces the release of pro-apoptotic factors, ultimately leading to caspase-dependent cell death [39]. Noise exposure promotes the nuclear translocation of endonuclease G, thereby inducing caspase-independent cell death [40].

Endogenous cellular processes, including autophagy, protect against NIHL by mitigating oxidative stress. Relatively low levels of oxidative stress induced by noise exposure at the TTS level activate autophagy and promote OHC survival. In contrast, excessive oxidative stress at the PTS level caused by noise exposure overwhelms the protective effects of autophagy, leading to OHC death and NIHL development of NIHL [41].

### 4.2. Calcium Stress and Glutamatergic Excitotoxicity

The cochlea converts mechanical sound signals into electrical signals, which are transmitted to the brain. In response to sound stimulation, calcium ions enter the cell, triggering calcium-induced calcium release (CICR) from the endoplasmic reticulum, leading to a significant increase in intracellular calcium concentration. This increase in calcium levels induces the release of neurotransmitters, primarily glutamate, from IHCs, facilitating the conversion of mechanical signals into electrical signals. Following acoustic overstimulation, calcium levels within hair cells, particularly in the OHCs, increase rapidly [42].

Calcium overload activates calcium-dependent enzymes such as calpains [43,44]. Over-activation of calpains, which degrade intracellular proteins and the cytoskeleton, causes structural collapse of cells via caspase- or cathepsin-associated pathways [45,46]. Furthermore, calcium overload has been suggested as a critical factor in glutamate excitotoxicity [47]. Glutamate is tightly regulated in the extracellular synaptic cleft, where its concentration is maintained at very low levels. However, excessive glutamate can destabilize the cysteine-glutamate transporter, leading to cysteine and glutathione depletion. Additionally, sustained activation of glutamate receptors on the postsynaptic membrane can result in excitotoxic damage to afferent nerve fibers [48].

### 4.3. Inflammatory Response

After noise exposure, activated resident macrophages undergo morphological changes, increase in number, and migrate to the hair cell region [49]. At the animal experimental level, noise exposure led to the upregulation of TLR4 expression in the cochlea. TLR4 is a pattern recognition receptor (PRR) and a transmembrane receptor expressed on the surface of immune cells. Activation of TLR4 induces the activation of NF-κB [50], which subsequently promotes the production of pro-inflammatory cytokines (e.g., TNF-α, IL-6, IL-1β). This inflammatory cascade contributes to the loss of outer hair cells and ribbon synapse damage [51,52]. Supporting this finding, a previous study reported that the inhibition of TLR4 signaling suppressed noise-induced macrophage activation, which consequently attenuated hair cell damage and led to a partial recovery of hearing function [53]. This indicates that the resident macrophages in the cochlea exhibit a pro-inflammatory phenotype and play a detrimental role in NIHL.

Excessive activation of the NF-κB pathway exacerbates inflammation in the inner ear and accelerates hearing loss; thus, inhibition of the NF-κB pathway may aid in recovery following NIHL [54]. Noise exposure induces activation of the NOD-like receptor protein 3 (NLRP3) inflammasome, triggering an inflammatory cascade. The interaction between the upregulation of NLRP3 expression and activation of NF-κB may further amplify inflammation within the cochlea [55]. 

Another cytokine implicated in NIHL is transforming growth factor-beta (TGF-β). Although the impact of the TGF-β family, a key regulator of immune and inflammatory responses [56], on NIHL remains incompletely understood, studies have reported that within the first 24 h following noise exposure, the gene expression of *Tgfb1* and *TgfbR1* is upregulated, while *Tgfb2* and *TgfbR2* are downregulated. These findings suggest that TGF-β1 is involved in the cochlear inflammatory response and may play a crucial role in the early inflammatory phase [57]. TGF-β1 is known to exert dual functions in the inflammatory response, exhibiting both pro-inflammatory and anti-inflammatory roles [58,59]. However, its overexpression can induce fibrosis, potentially leading to irreversible cochlear damage [60]. Inhibition of TGF-β1 has also been shown to exert a protective effect against NIHL [57].

In addition to the previously mentioned NF-κB pathway, several other inflammatory pathways are involved in NIHL [61]. The Mitogen-Activated Protein Kinase (MAPK) pathway plays a critical role in the cellular stress response and repair processes. Following NIHL, the MAPK pathway is activated, which induces inflammation and cell death [62]. KSR1 is a scaffold protein that facilitates the proximity of MAPK proteins, including BRAF, MEK1/2, and ERK1/2, thereby assisting their activation via a phosphorylation cascade in response to noise-induced damage. Deletion of KSR1 suppresses the MAPK phosphorylation cascade in cochlear cells following noise exposure, resulting in a protective effect on hearing [63]. Administration of the oral BRAF inhibitor dabrafenib confers protection against NIHL in KSR1 wild-type mice [63].

### 4.4. Endocochlear Potential (EP) Reduction

Noise exposure causes damage to the stria vascularis and spiral ligament, leading to hearing loss through changes in the endocochlear potential (EP). In the acute phase, degeneration of type II fibrocytes in the spiral ligament and edema in the stria vascularis were observed. In the chronic phase, extensive loss of type II fibrocytes, as well as degeneration of intermediate and marginal cells in the stria vascularis, were noted, resulting in a significant reduction in cell membrane surface area. In cases of TTS, no EP changes were observed, and EP changes do not appear to contribute significantly to stable PTS. However, EP reduction is involved in the acute threshold elevation and may subsequently contribute to the transition to PTS [64]. It has also been reported that disruption of the ion-trafficking system in cochlear spiral ligament fibrocytes is caused by a decrease in connexin levels and Na^+^-K^+^-ATPase activity [65]. Lateral wall pathology may promote hair cell loss and lead to the opening of the reticular lamina [66].

### 4.5. Ischemia

Cochlear blood flow decreases after noise exposure [67]. It has been reported that this may be due to the induction of potent vasoconstrictors, such as isoprostanes, in conjunction with the generation of ROS [68]. In cases of severe noise exposure to induce PTS, cochlear blood flow decreased more significantly than in TTS. Additionally, the diameter of the stria vascularis was decreased. Furthermore, the expression of genes involved in vasodilation decreased, whereas the expression of genes related to vasoconstriction increased [69]. Reperfusion injury caused by inner ear ischemia is thought to increase ROS levels and contribute to cochlear damage [70]. Multiple mechanisms have been elucidated to account for the generation of ROS during reperfusion. First, hypoxanthine, a byproduct of ATP catabolism, is metabolized to uric acid via xanthine by the action of xanthine oxidase, a process that concurrently generates ROS [71]. Second, NADPH oxidase (NOX), which is ordinarily maintained in an inactive state with its membrane-bound and cytosolic subunits spatially separated, becomes activate during ischemia–reperfusion through the assembly of these subunits, resulting in the production of ROS [72]. Third, ischemia impairs the mitochondrial electron transport chain, leading to a disrupted redox environment within the mitochondria [73]. Upon reperfusion, reactivation of the electron transport chain promotes electron leakage, which in turn facilitates the reduction in molecular oxygen to ROS. Fourth, endothelial nitric oxide synthase (eNOS), which typically synthesizes nitric oxide through coupling with tetrahydrobiopterin, undergoes uncoupling during ischemia–reperfusion [74]. In this state, the binding of 7,8-dihydrobiopterin to eNOS leads to the diversion of enzymatic activity toward the production of ROS rather than nitric oxide.

Furthermore, Vascular Endothelial Growth Factor (VEGF) levels increased after acoustic exposure, suggesting that endothelial stress and reduced blood flow led to cochlear hypoxia [75].

### 4.6. Changes in Auditory Cortex

Hearing loss caused by NIHL affects not only the cochlea but also the auditory cortex. Following acoustic exposure, the upregulated expression of inflammatory cytokines and activation of microglia are observed in the primary auditory cortex, indicating the presence of neuroinflammatory responses [76]. One proposed mechanism involves the increased expression of cytoplasmic receptors for advanced glycation end-products, which has been reported to enhance the expression of inflammatory genes and contribute to neuroinflammation [77].

Changes in spontaneous activity along the auditory pathway following NIHL have also been reported following NIHL [78,79]. In the early phase following NIHL (within a few days after noise exposure), glutamatergic excitatory activity in the brainstem decreases, whereas glycinergic and GABAergic inhibitory activities remain largely unchanged. In the late phase (approximately two months post-noise exposure), excitatory activity was suppressed, while inhibitory activity increased [80]. Moreover, in the auditory cortex, NIHL induces the deterioration of perineuronal nets, specialized extracellular matrix components that enwrap parvalbumin (PV)-expressing GABAergic interneurons [81]. These alterations may contribute to difficulties in speech comprehension as well as the development of tinnitus and hyperacusis [80]. Although the hippocampus lies outside the classical auditory pathway, it has been demonstrated that noise exposure chronically suppresses cell proliferation and neurogenesis within the hippocampus. Chronic exposure to noise leads to elevated levels of corticosterone, which binds to glucocorticoid hormone receptors abundantly expressed in hippocampal neurons, thereby inhibiting hippocampal cell proliferation and neurogenesis. Given the critical role of the hippocampus in spatial cognition, memory, and emotional regulation, noise-induced hearing loss may contribute to the impairment of these higher-order cognitive functions [82].

Furthermore, recent studies have suggested that plastic changes may also occur in brain regions beyond the auditory pathway, particularly in areas involved in higher-order cognitive functions, such as the prefrontal cortex [83].

## 5. Therapeutic Strategies: Potential of Antioxidant Therapy in NIHL

Clinically, glucocorticoids are the only approved medications for the treatment of NIHL. Glucocorticoids have been reported to be effective against NIHL through their anti-inflammatory effects. However, complete hearing recovery was not achieved, and in many cases, histological degeneration persisted [84], so other treatment options are also being sought. While current standard clinical interventions for NIHL include hearing aids and cochlear implants in such cases, these approaches mainly compensate for existing deficits rather than preventing or reversing damage. Furthermore, cochlear implants may not be effective in all patients, especially those with severe neural degeneration, making thorough pre-operative evaluation essential for surgical candidacy. In this context, molecular-based therapeutic strategies such as antioxidant therapy are being actively investigated in preclinical models as potential approaches to protect or restore cochlear structures, and may complement or enhance current clinical treatments in the future. Molecules involved in the aforementioned mechanisms can serve as potential therapeutic targets, and several drugs have shown therapeutic effects in preclinical animal models. Figure 2 depicts how representative therapeutic agents exert their effects.

### 5.1. MT (mito-TEMPO)

Mito-TEMPO (MT) is a mitochondria-targeted antioxidant that is effective in conditions associated with ROS, including renal fibrosis [85], ischemic brain injury [86], and acetaminophen-induced liver damage [87]. Animal studies have demonstrated that Mito-TEMPO is effective against NIHL. Mitochondrial DNA is susceptible to damage by ROS owing to a lack of protection from histone proteins and limited repair mechanisms. However, mitochondrial transcription factor A (TFAM), which is located in nuclear DNA, plays a crucial role in stabilizing and maintaining mitochondrial DNA. MT is believed to improve mitochondrial function by reducing oxidative stress and mitochondrial DNA damage in the inner ear as well as by restoring the interaction between TFAM and mtDNA, thus showing efficacy in NIHL [88].

### 5.2. 6-fluoro-9-methyl-pyridoindole (AC102)

6-fluoro-9-methyl-pyridoindole (AC102) is a pyridoindole derivative. Pyridoindoles are a group of organic compounds characterized by the fusion of a six-membered aromatic heterocyclic pyridine ring containing a single nitrogen atom with an indole structure comprising a benzene ring and a pyrrole ring. Notably, some pyridoindole derivatives have been reported to exhibit neuroregenerative potential in neurological disorders, such as Parkinson’s disease [89]. AC102 has been suggested to ameliorate NIHL by exerting neuroprotective effects through the reduction in ROS and enhancing synaptic plasticity via increased ATP production [26].

### 5.3. Calpain Inhibitor (MDL-28170)

The calpain inhibitor MDL-28170 is a potent, cell-permeable inhibitor of calpain-1 and calpain-2 and is capable of crossing the blood–brain barrier (BBB) [90]. Noise exposure activates calpain, which inhibits downstream PI3K/Akt signaling, thereby promoting apoptosis and leading to inner ear cell death. However, treatment with the calpain inhibitor, MDL-28170, enhanced the PI3K/Akt signaling pathway, contributing to the prevention of NIHL [91]. Reports have demonstrated that direct injection of the calpain inhibitor MDL-28170 into the scala tympani effectively attenuates NIHL [92].

In addition, Qter, a synthetic analog of the endogenous antioxidant coenzyme Q10, and avenanthramide-C, a natural flavonoid purified from oats, have been reported at the experimental animal level to be beneficial for hearing protection against NIHL through their anti-inflammatory and antioxidant effects [93,94].

## 6. Suggesting New Avenues for Future Research

The mechanisms by which oxidative stress induced by noise exposure leads to NIHL have been gradually elucidated, and various candidate therapeutic agents have been reported. However, there remains significant difficulty in the clinical implementation of these drugs. One major obstacle is the presence of the blood–labyrinth barrier, which tightly regulates the exchange of substances between the bloodstream and the cochlea [95]. Due to this barrier, systemically administered drugs, whether oral or intravenous, often fail to reach therapeutic concentrations within the cochlea, thereby limiting their efficacy. One potential strategy to overcome this challenge is the development of nanoparticle-based drug delivery systems that can either cross the blood–labyrinth barrier or be locally administered into the middle ear, allowing drugs to pass through the round or oval window membranes into the cochlea [96]. Specifically, nanosystems utilizing polyethylene glycol (PEG)-coated polylactic acid (PLA) nanoparticles and zeolitic imidazolate framework (ZIF)-90 nanoparticles have been developed, and their effectiveness in facilitating drug delivery to the inner ear has been demonstrated [97,98]. Another promising approach involves the use of viral vectors as carriers for therapeutic agents, delivered via the posterior semicircular canal [99] or through the round window membrane [100]. Using superparamagnetic iron oxide nanoparticles (SPIONs) and a recombinant adeno-associated virus vector (AAV2), minimally invasive magnetic targeting of brain-derived neurotrophic factor (BDNF) gene therapy to the inner ear has been reported to potentially reverse cochlear synaptopathy following NIHL [101]. Although animal studies have made progress in identifying viral types with high transduction efficiency [102,103], clinical translation of such methods is hampered by concerns regarding invasiveness and safety. In parallel, hearing aids remain the primary clinical solution for managing NIHL in affected individuals. Therefore, both established technological interventions and emerging pharmacological strategies are critical in addressing the multifaceted challenges of NIHL.

## 7. Conclusions

NIHL is the second leading cause of sensorineural hearing loss, and the number of affected individuals is expected to increase due to changes in societal conditions. However, no effective treatment has yet been established, making the development of therapeutic strategies a significant public health challenge. Although the mechanisms underlying NIHL have not been fully elucidated, increasing evidence sheds light on the pathophysiological processes involved. Consequently, targeting the molecular factors responsible for NIHL has emerged as a promising approach for the development of effective treatments.

## Figures and Tables

**Figure 1 ijms-26-04927-f001:**
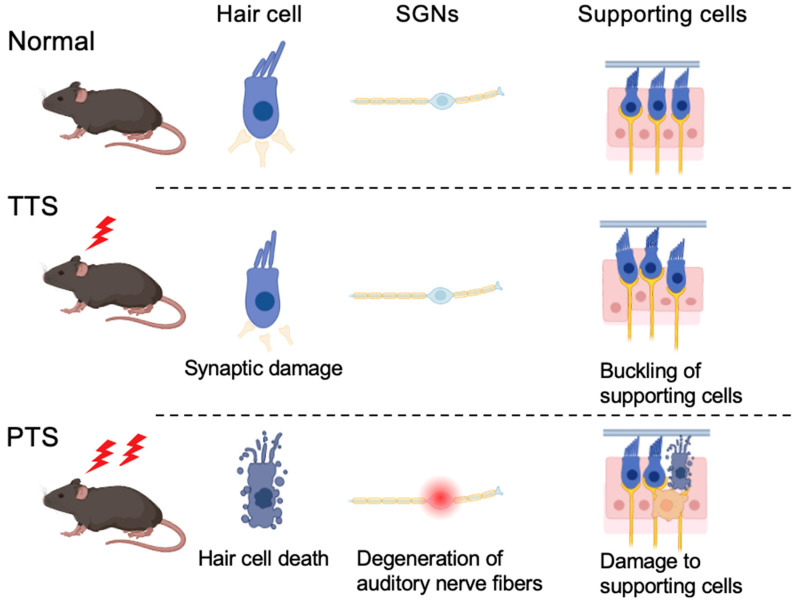
Histological differences between temporary threshold shift (TTS) and permanent threshold shift (PTS). In TTS, the hair cells remain intact; however, ribbon synapses are damaged and become detached. Additionally, supporting cells undergo buckling, causing some outer hair cells to become detached from the tectorial membrane. When noise exposure is more intense or repeated, hair cells’ and supporting cells’ death and the auditory nerve’s degeneration can produce an irreversible condition, leading to PTS.

**Figure 2 ijms-26-04927-f002:**
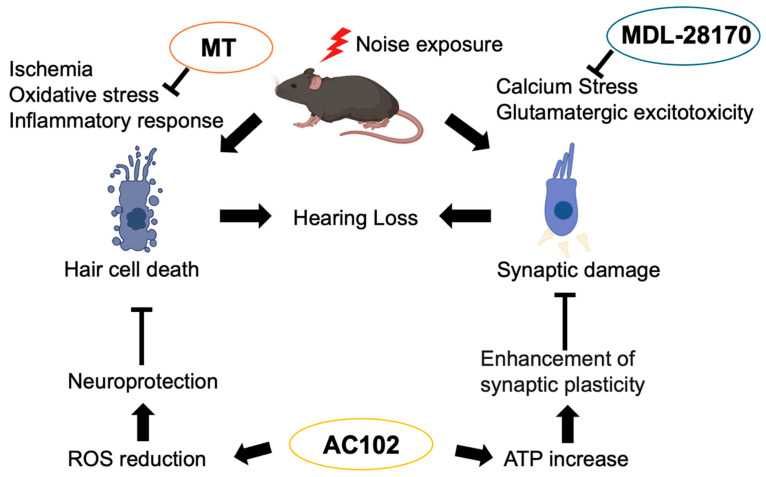
Molecular mechanisms of noise-induced hearing loss and potential therapeutic targets. The pathogenesis of noise-induced hearing loss (NIHL) involves multiple mechanisms, including oxidative stress, inflammatory responses, and ischemia-induced hair cell death, as well as calcium stress and glutamatergic excitotoxicity, which contribute to synaptic damage. Several pharmacological agents have been proposed as potential therapeutic candidates for NIHL, including mito-TEMPO (MT), a mitochondria-targeted superoxide dismutase (SOD) mimetic that mitigates oxidative stress; 6-fluoro-9-methyl-pyridoindole (AC102), which has demonstrated neuroprotective effects; and calpain inhibitor MDL-28170, which suppresses calpain-mediated apoptosis and preserves cochlear integrity. In the figure, arrows indicate interactions between components. Activation arrows (→) represent stimulatory effects, while inhibitory arrows (⊣) represent suppressive effects.

## Data Availability

Not applicable.

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
