# Peer review of "Noise-Induced Hearing Loss: Overview and Future Prospects for Research on Oxidative Stress"

_ijms, 2025, doi:10.3390/ijms26104927_

Round 1

Reviewer 1 Report

Comments and Suggestions for Authors

The manuscript with title "Noise-Induced Hearing Loss: Overview and Future Prospectives on Oxidative Stress" could be an interesting review if the following information will be added:

-Types of the reactive oxygen species and of the nitrogen species that were seen to develop in experimental models. The title of this manuscript obliges to more data related to oxidative stress in this disease. These parameters of oxidative stress can be added as text or as a table.

-The antioxidants that were studied in the experimental models. The Therapeutic strategies, part of the manuscript, presents only 3 drugs. These are the only type of medication mentioned in literature? Other types of antioxidants must be mentioned, natural and synthetic, with their effects.

-The "Suggesting new avenues for future research" is presented only as a general idea. More details related to nanoparticles as drug delivery, viral types, must be added. 

-The Figures are not mentioned in the text.

Author Response

The manuscript with title "Noise-Induced Hearing Loss: Overview and Future Prospectives on Oxidative Stress" could be an interesting review if the following information will be added:

-Types of the reactive oxygen species and of the nitrogen species that were seen to develop in experimental models. The title of this manuscript obliges to more data related to oxidative stress in this disease. These parameters of oxidative stress can be added as text or as a table.

Thank you for your appropriate comments. In accordance with your comments, we revised the manuscript as shown below. L139 "Among the ROS, superoxide anion (O₂⁻), hydroxyl radical (·OH), and hydrogen peroxide (H₂O₂) have been frequently reported to accumulate in the cochlea following acoustic overstimulation. Similarly, reactive nitrogen species such as nitric oxide (NO) and peroxynitrite (ONOO⁻) are also generated in response to noise exposure."

-The antioxidants that were studied in the experimental models. The Therapeutic strategies, part of the manuscript, presents only 3 drugs. These are the only type of medication mentioned in literature? Other types of antioxidants must be mentioned, natural and synthetic, with their effects.

Thank you for your appropriate comments. In accordance with your comments, we revised the manuscript as shown below. L290 "Clinically, glucocorticoids are the only approved medications for the treatment of NIHL. Glucocorticoids have been reported to be effective against NIHL through their anti-inflammatory effects. However, complete hearing recovery was not achieved, and in many cases, histological degeneration persisted, so that other treatment options area also being sought."   L334 "In addition, Qter, a synthetic analog of the endogenous antioxidant coenzyme Q10, and avenanthramide-C, a natural flavonoid purified from oats, have been reported at the experimental animal level to be benefical for hearing protection against NIHL through their anti-inflammatory and antioxidant effects."

-The "Suggesting new avenues for future research" is presented only as a general idea. More details related to nanoparticles as drug delivery, viral types, must be added. 

Thank you for your appropriate comments. In accordance with your comments, we revised the manuscript as shown below. L360 "Specifically, nanosystems utilizing polyethylene glycol (PEG)-coated polylactic acid (PLA) nanoparticles and zeolitic imidazolate framework (ZIF)-90 nanoparticles have been developed, and their effectiveness in facilitating drug delivery to the inner ear has been demonstrated." L365 "Using superparamagnetic iron oxide nanoparticles (SPIONs) and a recombinant adeno-associated virus vector (AAV2), minimally invasive magnetic targeting of brain-derived neurotrophic factor (BDNF) gene therapy to the inner ear has been reported to potentially reverse cochlear synaptopathy following NIHL."

-The Figures are not mentioned in the text.

Thank you for your appropriate comments. In accordance with your comments, we revised the manuscript as shown below.

L96

"Figure 1 shows the histological changes associated with PTS and TTS which have been identified:"

L303

"Figure 2 depicts how representative therapeutic agents exert their effects."

Reviewer 2 Report

Comments and Suggestions for Authors

Dear authors,

Thank you for submitting your manuscript. Here are my recommendations to optimize it:

40ff: Please re-write.

74f: Please add a reference.

77ff: Animal model? Please clarify.

148f: Please re-write.

252: Please define VEGF.

282ff: This title and its sub-section's content sounds misleading as , e.g., hearing aids and cochlear implants are the established ways to help humans with hearing loss including NIHL. Please clarify and include a hint that in some cases of deafness even cochlear implants can't help. (That's why a pre-operative (optimal: objective) test is recommended to verfiy the patients candidacy for the surgery.) Though the journal's scope is on molecular science this should be stated.

339ff: The role of hearing aids in treatment is not also important. Hearing aid is the treatment/solution today.

Author Response

Thank you for submitting your manuscript. Here are my recommendations to optimize it:

40ff: Please re-write.

Thank you for your appropriate comments. In accordance with your comments, we revised the manuscript as shown below.

L42 "The advancement of audio technology and the rapid expansion of the music industry have contributed to a growing trend among adolescents and young adults of voluntarily listening to music at high volumes for extended periods. Such prolonged exposure to loud sounds significantly increases the risk of NIHL, raising public health concerns about a potential rise in the prevalence of NIHL in the future."

74f: Please add a reference.

Thank you for your comment. We have added the reference [15] and [16] to support the revised statement.

77ff: Animal model? Please clarify.

Thank you for your appropriate comments. In accordance with your comments, we revised the manuscript as shown below.

L79

"Animal studies indicate that immediate threshold shifts of approximately 50 dB or less may result in complete recovery, whereas shifts exceeding this level are more likely to result in permanent hearing loss."

148f: Please re-write.

Thank you for your appropriate comments. In accordance with your comments, we revised the manuscript as shown below.

L154

"In addition, damage to supporting cells, the stria vascularis, and auditory nerve cells in the inner ear can ultimately lead to hearing loss."

252: Please define VEGF.

Thank you for your appropriate comments. In accordance with your comments, we revised the manuscript as shown below.

L258

"Furthermore, Vascular Endotherial Growth Factor (VEGF) levels increased after acoustic exposure, suggesting that endothelial stress and reduced blood flow led to cochlear hypoxia."

282ff: This title and its sub-section's content sounds misleading as , e.g., hearing aids and cochlear implants are the established ways to help humans with hearing loss including NIHL. Please clarify and include a hint that in some cases of deafness even cochlear implants can't help. (That's why a pre-operative (optimal: objective) test is recommended to verfiy the patients candidacy for the surgery.) Though the journal's scope is on molecular science this should be stated.

Thank you for your appropriate comments. In accordance with your comments, we revised the manuscript as shown below.

L289

"Therapeutic Strategies: Potential of Antioxidant Therapy in NIHL"

L294

"While current standard clinical intervensions for NIHL include hearing aids and cochlear implants in such cases, these approaches mainly compensate for existing deficits rather than preventing or reversing damage. Furthermore, cochlear implants may not be effective in all patients, especially those with severe neural degeneration, making thorough pre-operative evaluation essential for surgical candidacy. In this context, molecular-based therapeutic strategies such as antioxidant therapy are being actively investigated in preclinical models as potential approaches to protect or restore cochlear structures, and may complement or enhance current clinical treatments in the future."

339ff: The role of hearing aids in treatment is not also important. Hearing aid is the treatment/solution today.

Thank you for your appropriate comments. In accordance with your comments, we revised the manuscript as shown below.

L371

In parallel, hearing aids remain the primary clinical solution for managing NIHL in affected individuals. Therefore, both established technological interventions and emerging pharmacological strategies are critical in addressing the multifaceted challenges of NIHL."

Round 2

Reviewer 1 Report

Comments and Suggestions for Authors

Dear Authors,

Please revise again the manuscript. Small errors occured, like in line 258 where "endotherial" is written.

After the accurate revision,  the manuscript can be published.